# The effects of antibiotics and illness on gut microbial composition in the fawn-footed mosaic-tailed rat (*Melomys cervinipes*)

**Tasmin L. Rymer** [1,2,3]*, **Neville Pillay** [3]

**1** College of Science and Engineering, James Cook University, Cairns, Queensland, Australia, **2** Centre for Tropical Environmental and Sustainability Sciences, James Cook University, Queensland, Australia, **3** School of Animal, Plant and Environmental Sciences, University of the Witwatersrand, Johannesburg, South Africa

\* tasmin.rymer@jcu.edu.au

**Data Availability Statement:** All relevant data are within the manuscript and its Supporting Information files.

**Funding:** The author(s) received no specific funding for this work.

## Abstract

The gut microbiota are critical for maintaining the health and physiological function of individuals. However, illness and treatment with antibiotics can disrupt bacterial community composition, the consequences of which are largely unknown in wild animals. In this study, we described and quantified the changes in bacterial community composition in response to illness and treatment with antibiotics in a native Australian rodent, the fawn-footed mosaic-tailed rat (*Melomys cervinipes*). We collected faecal samples during an undiagnosed illness outbreak in a captive colony of animals, and again at least one year later, and quantified the microbiome at each time point using 16s ribosomal rRNA gene sequencing. Gut bacterial composition was quantified at different taxonomic levels, up to family. Gut bacterial composition changed between time periods, indicating that illness, treatment with antibiotics, or a combination affects bacterial communities. While some bacterial groups increased in abundance, others decreased, suggesting differential effects and possible co-adapted and synergistic interactions. Our findings provide a greater understanding of the dynamic nature of the gut microbiome of a native Australian rodent species and provides insights into the management and ethical well-being of animals kept under captive conditions.

## Introduction

The mammalian intestinal tract houses an immense diversity of microbial organisms (upwards of 100 trillion [1, 2]). These organisms, collectively known as the gut microbiota, help to maintain a balance between the microecological environment and host physiological functioning, including immunity [3–5], digestion [6], nutrition [7], metabolic function [8] and defence against pathogens [9]. Thus, they are critical for the overall health [10, 11] and growth [6] of the host. Importantly, the gut microbiome is strongly influenced by multiple external factors, including diet [12], conspecifics [13] and pharmacological drugs, such as antibiotics [14].

Antibiotics can have both positive and negative effects on gut microbial composition due to direct differential species responses (e.g., Gram-positive bacteria have a permeable cell wall

**Competing interests:** The authors have declared that no competing interests exist.

that generally does not restrict the penetration of antibiotics [15]) and indirect effects (e.g., interacting effects, where a reduction or elimination of one species may allow another species to thrive due to a reduction in competition for space and nutrients [16]). Consequently, changes in microbial composition impact host biological functioning. For example, the bacterial family Muribaculaceae is known to decrease in abundance with antibiotic treatment [11], and decreased abundance of Muribaculaceae is associated with anxiety and depressive-like behaviours in C57BL/6 laboratory mice [17]. In contrast, the bacterial family Akkermansiacae increases in abundance with antibiotic treatment [11]. These bacteria provide protection from intestinal inflammation [18–20] and gut barrier impairment [21], and increase in response to physiological stress to help rebalance the gut microbiota [17].

While the effects of antibiotics on the gut microbiome have principally focused on humans and laboratory animals, understanding how antibiotics impact the gut microbiome also has relevance for wildlife. Trevelline et al. [22] suggested that an understanding of the gut microbiome could have significant relevance for understanding host-microbe coevolution in wild animals, while Chong et al. [23] suggested that it could also offer important insights for conservation. There are two considerations. Firstly, increasing anthropogenic impacts place direct pressure on species through increased risk of extinction, resulting in threatened species often being confined to captivity for management purposes [23]. The captive management of animals could result in the direct administration of antibiotics in response to illness [23]. While antibiotics are essential in this context, improving the lives and health of animals [24], they would have a corresponding effect on the gut microbiome, which could be positive or negative. Secondly, antibiotic usage by humans can have collateral effects on wildlife [24]. Anthropogenic impacts may have an indirect effect on species through environmental contamination, where antibiotics enter ecosystems via waterways [25] and these effects can further be carried up food chains [26]. Investigation of the effects of antibiotics on the gut microbiome of wild species thus provide valuable information that could be useful for their management and conservation.

Fawn-footed mosaic-tailed rats (*Melomys cervinipes*) are medium-sized nocturnal murid rodents endemic to Australia [27]. In 2018, individuals in a captive colony of mosaic-tailed rats presented with an undiagnosed illness (results from blood tests, nasal swabs, stool samples and x-rays provided no definitive diagnosis as to the cause), giving us an opportunity to explore the effects of antibiotic treatment and illness on the gut microbiota. Symptoms included diarrhoea, lethargy, inappetence and weight loss. Consultation with a local veterinarian saw affected animals treated with Bactrim (a combination of two antibiotics: sulfamethoxazole and trimethoprim). Animals responded favourably and recovered to full health on this treatment. Healthy animals have a vigorous appetite (including active begging behaviours), bright, wide-open eyes, maintenance of a good weight (70–80 g [27]), a well-groomed pelage and demonstrate species-typical behaviours, including climbing [27].

The aim of the study was therefore to describe and quantify the changes in bacterial community composition in response to illness and treatment with antibiotics. As this was an opportunistic study, we made no *a priori* predictions on the direction of the effects of antibiotics on the gut microbial composition of mosaic-tailed rats; however, we did expect to observe differences in gut microbial composition because of either illness, treatment with antibiotics, or both.

## Materials and methods

### Ethical note

The research complied with the Australian Code for the Care and Use of Animals for Scientific Purposes [28] and the ABS/ASAB guidelines for the ethical treatment of animals [29]. The

Animal Ethics Screening Committee of James Cook University approved housing and husbandry of animals (clearance number: A2539). The wild-caught individuals used in this study were originally trapped with permission from the Department of Science (permit numbers: WISP14530814 and WITK14530914). Individuals were observed daily and received behavioural enrichment (scattered seeds to stimulate foraging, platforms and sticks for climbing, and cardboard rolls and wooden blocks for chewing). Individuals were weighed every two weeks to monitor health. The onset of the illness was sudden and without obvious cause. When the first four individuals started showing symptoms, they were immediately transported to a veterinarian for assessment. Unfortunately, these four individuals died prior to receiving treatment. Twenty-three individuals that were treated with antibiotics recovered completely. No further illnesses have occurred in this colony since. No animals were euthanised for the purposes of this study. All deaths were natural and a consequence of the undiagnosed illness.

## Subjects

Mosaic-tailed rats used in this study originated from a combination of wild-caught and F1 captive-born individuals (wild-caught: n = 15; captive-born: n = 14). All individuals were sexually mature and had been kept in captive conditions for at least 12 months before the study. For details on the general husbandry of mosaic-tailed rats, see Rowell & Rymer [30]. Briefly, mosaic-tailed rats were housed individually in wire-frame cages with wood shavings for bedding, and a cylindrical plastic nest box, hay, and paper towel for nesting material. Environmental enrichment items were provided. Each individual was fed ± 5 g of mixed seeds and rodent chow (Vetafarm Origins), and ± 5 g of fruits or vegetables (e.g., apple, cucumber) daily. Water was available *ad libitum*.

## Sample collection and preparation

While 23 individuals were treated with antibiotics, faecal samples were only collected from 14 individuals (6 males and 8 females; individuals randomly selected for each sex from the available infected individuals) due to financial limitations associated with sequencing (designated TREATMENT). NP was blind to the group allocation. Faecal samples were collected fresh during routine husbandry during the period of antibiotic treatment, with the intention of collecting at least 1 g of faecal samples per individual. We also collected faecal samples from the same treated individuals (designated POST-TREATMENT) over 1 year later. Because this study was opportunistic, there were no incidences of sick individuals not receiving antibiotics and later recovering, nor were there incidences of healthy animals receiving antibiotics (i.e., no controls). Faeces were initially frozen at -20˚C and later sent to the Australian Centre for Ecogenomics (ACE; https://ecogenomic.org/) sequencing laboratory for DNA extraction and sequencing.

## DNA extraction protocol

DNA was extracted by ACE using the following protocols. The DNA was first extracted from 40–200 mg of the sample. The sample was bead beaten using 0.1–0.15 mm Zirconia/silica beads (BioSpec Products #11079101z) on a Powerlyser 24 homogenizer (Mo-Bio #13155) as per the manufacturer's instructions. Thereafter, 1.2 ml of Lysis buffer (Perkin Elmer Cat No #CMG-1076) was added to each tube, and tubes were then vortexed. Thereafter, 30 µl of Proteinase K (Perkin Elmer Cat No #CMG-820) was added to each sample, samples were vortexed, and were then incubated at 70˚C for 10 mins. Samples were incubated at 95˚C for a further 5 mins and then processed on a MoBio Powerlyzer for 5 mins at 2000 rpm. Samples were then centrifuged for 1 min at 10000g. For each sample, 800 µl of supernatant was

transferred to a deep well on a 96 well plate. DNA extraction was performed using a Chemagic™ 360 instrument (#2024–0020) following the manufacturer's protocol for Purification for Human Faeces using 75 μl specially washed magnetic beads (Perkin Elmer Cat No #CMG-1076) and eluted into 50 μl of buffer. The DNA concentration was measured using a Qubit high sensitivity assay (ThermoFisher Scientific; Qubit 3.0 and #Q32854). This was then adjusted to a concentration of 5 ng/ul. The extracted DNA was of sufficient quality to proceed without requiring further dilution or clean up.

## PCR amplification and amplicon sequencing protocols

The 16s rRNA gene encompassing the V6 to V8 regions was targeted using the 926F (5'-AA ACTYAAAKGAATTGRCGG -3') and 1392wR (5'-ACGGGCGGTGWGTRC-3') primers [31]. These were modified to contain Illumina specific adapter sequences (926F:5'TCGTCGGCAGCGTCAGATGTGTATAAGAGACAGAAACTYAAAKGAATTGRCGG3' and 1392wR: 5'GTCTCGTGGGCTCGGGTCTCGTGGGCTCGGAGATGTGTAT AAGAGACA GACGGGCGGTGWGTRC3'). The small subunit ribosomal RNA of eukaryotes (18s) and prokaryotes (16s), specifically the V6, V7 and V8 regions, were amplified by the universal primer pair Univ_SSU_926F-1392wR.

The 16s library was prepared using the workflow outlined by Illumina (#15044223 Rev.B). PCR productions of ~466bp were first amplified according to the defined workflow, with an alteration in polymerase used to substitute NEBNext® Ultra™ II Q5® Mastermix (New England Biolabs #M0544) in standard PCR conditions. Agencourt AMPure XP beads (Beckman Coulter) were used to purify the resulting PCR amplicons. The Illumina Nextera XT 384 sample Index Kit A-D (Illumina FC-131-1002) was then used to index the purified DNA with unique 8 bp barcodes in standard PCR conditions with NEBNext® Ultra™ II Q5® Mastermix. Following the manufacturer's protocol, the indexed amplicons were then pooled in equimolar concentrations and sequenced on the MiSeq Sequencing System (Illumina) using paired end sequencing with V3 300 bp chemistry.

The following control reactions were included in the amplicon library construction and sequencing: 1) Positive amplification control to monitor bias in the amplicon library construction. This was performed from a known mock community. 2) Negative amplification control to monitor contamination in library construction. This was performed from a like processed reagent control; 3) Single well empty chamber controls to monitor cross contamination within the library preparation. This was performed within processing plates; and 4) Negative index positions between runs, designated as in line controls, to monitor for run-to-run bleed through.

The passing quality control of the resulting sequences was determined as 10,000 raw reads per sample prior to data processing and passing quality control metrics in line with Illumina supplied reagent metrics of overall Q30 for 600bp reads of > 70%.

## Sample analysis

ACE provides the relative abundance of different bacterial groups from the taxonomic domain through to species, where possible. We calculated the relative abundance of different bacterial taxa within each taxonomic rank (where there was sufficient data to allow statistical analyses) as a percentage of the overall abundance.

## Statistical analysis

Statistical analyses were performed using RStudio (version 2022.02.3; https://www.rproject. org; R version 4.2.1, https://cran.rstudio.com). Data from all animals were used. No data points

were excluded from any analyses. Thus, data in all models included all individuals (n = 14). While the total number of reads was different between time periods (during antibiotic treatment and after), a comparison of the proportions at the Domain level for Archaea and Eukarya showed no significant difference (Wilcoxon rank sum test with continuity correction: Archaea: W = 382, p = 0.192; Eukarya: W = 382, p = 0.160), suggesting that normalizing the data using proportions is appropriate [32]. The model-level significance for all models was set at α = 0.05.

Due to the extensive size of the microbiome generally, we do not report all patterns at all taxonomic levels. Rather, we systematically worked from the level of Domain, exploring only those groups accounting for more than 1% (in either the TREATMENT or the POST-TREATMENT group) of the bacterial diversity at each taxonomic level. Only significant statistics are reported in text. All remaining results are reported in S1 Table.

We first used separate principal components analyses (PCA; corrplot package [33]) at each taxonomic level (i.e., phylum, class, etc.), except Domain, incorporating the groups accounting for more than 1% of the diversity, to reduce the number of predictors, and to discern potential ecological relationships between different bacterial groups in relation to treatment. We only included a principal component (PC) in later analyses if the eigen value was above 1. We report the variance (alone and combined) for included PCs.

Thereafter, for each taxonomic level (except Domain), we tested for treatment (TREATMENT vs POST-TREATMENT), sex (male or female) and birth origin (captive-born or wild-caught) effects on the magnitude of the PCs using separate rank-based non-parametric analyses for longitudinal data (F2-LD-F1 design, nparLD package [34]). We included the random effect of individual identity as a subject in these models. These analyses offer a robust framework for non-continuous variables, small sample sizes and skewed data [34]. Only significant effects are noted in the text, but all results are presented in S1 Table. Cohen's *d* effect sizes were calculated for the main effects only using the effectsize package [35] and are presented with confidence intervals in brackets.

We then ran individual rank-based non-parametric analyses for longitudinal data (F2-LD-F1 design) to determine which bacterial groups might be the main contributors of patterns of variation within each PC. We only report statistical information for bacterial groups where the factor under scrutiny was significant (e.g., when analysing phyla, if there was a treatment effect for PC1, we systematically ran models for each bacterial phylum contributing the most to that PC, and report only those models that showed a treatment effect; main text and S1 Table).

We used the phyloseq package [36] to graphically display the alpha diversity of the gut bacterial community from each time period. We also compared overall bacterial community abundance for each time period using paired t-tests. Finally, we identified species of particular pathogenic interest unique to each time period, and then used the KEGG database (Kyoto Encyclopedia of Genes and Genomes; (https://www.kegg.jp/kegg/) to explore potential functional pathways. We used FunGene (the functional gene pipeline & repository; http://fungene.cme.msu.edu/) to briefly explore some relationships with known antibiotic resistance genes [37].

## Results

### Domain

There was a significant effect of treatment on the abundance of Bacteria (ATS = 41.17, df = 1; $p < 0.001$; $d$ = 1.52 [0.66, 2.35]). The period of antibiotic treatment resulted in a significantly lower abundance of Bacteria (mean ± SE: TREATMENT: 84.31 ± 3.05%; POST-TREATMENT: 97.82 ± 1.41%; Fig 1). There were no other significant effects or interactions on the abundance of Bacteria in the microbiome (S1 Table).

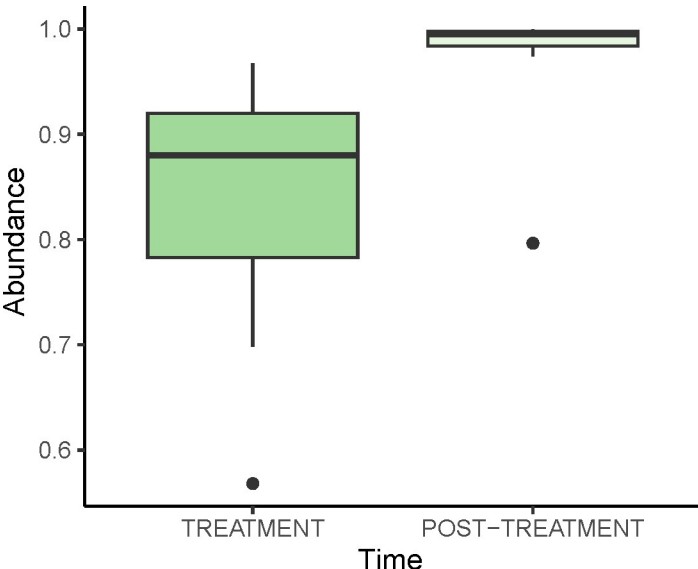

**Fig 1. Box and whisker plot of Domain Bacteria (%) in fawn-footed mosaic-tailed rat (*Melomys cervinipes*) faecal samples.** For rats treated with antibiotics (TREATMENT) and more than one year later (POST-TREATMENT).

## Phylum

12 bacterial phyla were common in both time periods. Of these, six phyla collectively accounted for more than 99% of the bacterial diversity, regardless of the period of time (TREATMENT: Bacteroidota: 42.23 ± 5.06%; Cyanobacteria: 2.53 ± 1.01%; Bacillota: 35.32 ± 1.72%; Fusobacteria: 1.60 ± 1.25%; Pseudomonadota: 9.13 ± 3.39%; Verrucomicrobiota: 8.52 ± 2.78%; POST-TREATMENT: Bacteroidota: 48.39 ± 2.41%; Cyanobacteria: 0.68 ± 0.16%; Bacillota: 44.68 ± 3.38%; Fusobacteriota: 0.12 ± 0.12%; Pseudomonadota: 1.94 ± 0.50%; Verrucomicrobiota: 3.54 ± 1.23%).

For bacterial phyla, the first and second PCs collectively explained 66.29% of the variance (S2 Table). For PC1 (hereafter PC_Phylum1), Verrucomicrobiota contributed the most to the variance (31%), followed by Bacteroidota (25%), Fusobacteriota (23%) and Pseudomonadota (17%). Verrucomicrobiota, Fusobacteriota and Pseudomonadota were all positively correlated with each other, and all were negatively correlated with Bacteroidota (S3 Table). For PC2 (hereafter PC_Phylum2), Bacillota contributed the most to the variance (55%), followed by the Cyanobacteria (28%). However, there was no significant correlation between these two bacterial phyla., and the abundance of Cyanobacteria was not correlated with any other phylum. The Bacillota was significantly negatively correlated with all PC_Phylum1 phyla (S3 Table).

While there was no significant treatment effect for PC_Phylum1 (ATS = 0.13; df = 1; $p = 0.715$; $d = 0.61$ [-0.15, 1.37]; Fig 2A), there was a significant treatment effect for Pseudomonadota (ATS = 18.62; df = 1; $p < 0.001$; $d = -0.79$ [-1.56, -0.02]), which showed a significantly higher abundance during the period of antibiotic treatment (Fig 2B). There was also a significant birth * treatment effect on the abundance of Pseudomonadota (ATS = 14.60; df = 1; $p < 0.001$), with captive-born individuals showing a significantly lower abundance of Pseudomonadota during the period following treatment with antibiotics and illness. Furthermore, there was a significant treatment effect for PC_Phylum2 (ATS = 9.51; df = 1; $p = 0.002$; $d = -0.85$ [-1.62, -0.06]; Fig 2A), which was likely associated with the abundance of Bacillota (ATS = 10.04; df = 1; $p = 0.002$; $d = 0.93$ [0.14, 1.71]). There was a significantly lower

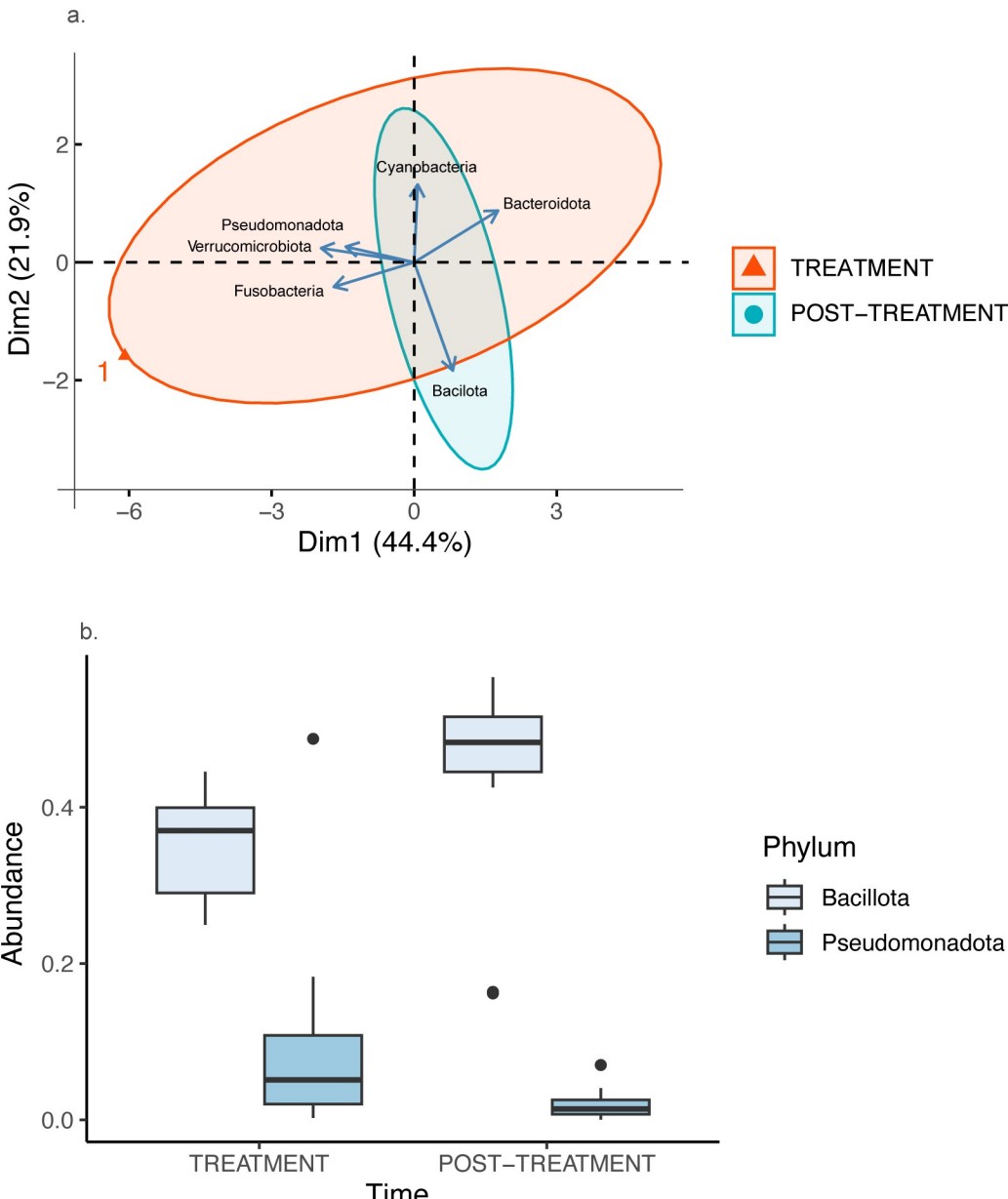

**Fig 2.** (a) Principal components analysis and (b) Box and whisker plot of two Bacterial Phyla in fawn-footed mosaic-tailed rat (*Melomys cervinipes*) faecal samples. Principal components analysis shows the first two principal components of 6 bacterial phyla for rats treated with antibiotics (red) and more than one year later (blue). Box and whisker plot of Phyla Bacillota (%) and Peusodomonadota (%) for both treatments.

abundance of Bacillota during the period of antibiotic treatment (Fig 2B). There were no other significant effects (S1 Table).

## Class

14 bacterial classes were common in both time periods. Of these, eight accounted for more than 93% of the bacterial diversity, regardless of the period of time (TREATMENT: Alphaproteobacteria: 2.05 ± 0.89%; Bacilli: 6.03 ± 1.66%; Bacteroidia: 42.23 ± 5.06%; Clostridia:

24.43 ± 1.56%; Fusobacteriia: 1.60 ± 1.25%; Gammaproteobacteria: 7.09 ± 3.44%; Negativicutes: 1.21 ± 0.35%; Verrucomicrobiae: 8.52 ± 2.78%; POST-TREATMENT: Alphaproteobacteria: 0.68 ± 0.36%; Bacilli: 11.97 ± 1.60%; Bacteroidia: 48.39 ± 2.41%; Clostridia: 32.49 ± 2.80%; Fusobacteriia: 0.12 ± 0.12%; Gammaproteobacteria: 1.26 ± 0.29%; Negativicutes: 0.18 ± 0.17%; Verrucomicrobiae: 3.54 ± 1.23%).

For bacterial classes, the first and second PCs collectively explained 60.37% of the variance (S2 Table). For PC1 (hereafter PC_Class1), Verrucomicrobiae contributed the most to the variance (31%), followed by Bacteroidia (23%), Fusobacteriia (21%) and Gammaproteobacteria (16%). Verrucomicrobiae, Fusobacteriia and Gammaproteobacteria were all positively correlated with each other, and all were negatively correlated with Bacteroidia, although the relationship was only significant between Bacteroidia and Verrucomicrobiae (S3 Table). For PC2 (hereafter PC_Class2), Negativicutes contributed the most to the variance (26%), followed by the Clostridia (20%), Alphaproteobacteria (19%) and the Bacilli (18%). The Negativicutes were significantly negatively correlated with both the Bacilli and Clostridia, which were positively correlated with each other (S3 Table). There were no other significant correlations observed (S3 Table).

There was no significant treatment effect for PC_Class1 (ATS = 0.11; df = 1; $p$ = 0.744; Fig 3A; $d$ = 0.55 [-0.21, 1.30]). However, there was a significant treatment effect for PC_Class2 (ATS = 10.26; df = 1; $p$ = 0.001; $d$ = 0.97 [0.18, 1.75]; Fig 3A), which was likely associated with the abundance of Bacilli (ATS = 6.96; df = 1; $p$ = 0.008; $d$ = -1.05 [-1.83, -0.25]), Clostridia (ATS = 8.49; df = 1; $p$ = 0.004; $d$ = 0.95 [0.16, 1.73]) and Negativicutes (ATS = 9.68; df = 1; $p$ = 0.002; $d$ = -1.00 [-1.78, -0.21]). There was a significantly lower abundance of Bacilli and Clostridia during the period of antibiotic treatment, whereas there was a significantly higher abundance of Negativicutes during this period (Fig 3B). There were no other significant effects (S1 Table).

## Order

27 bacterial orders were common across both time periods. Of these, 10 accounted for more than 63% of the bacterial diversity, regardless of the period of time (TREATMENT: Bacteroidales: 42.22 ± 5.06%; Enterobacterales: 4.46 ± 3.36%; Erysipelotrichales: 3.65 ± 0.70%; Eubacteriales: 24.42 ± 1.56%; Fusobacteriales: 1.60 ± 1.25%; Gastranaerophilales: 2.50 ± 1.02%; Lactobacillales: 5.97 ± 1.67%; Rhodospirillales: 2.02 ± 0.89%; Selenomonadales: 1.21 ± 0.35%; Verrucomicrobiales: 8.52 ± 2.78%; POST-TREATMENT: Bacteroidales: 48.26 ± 2.42%; Enterobacterales: 0.12 ± 0.12%; Erysipelotrichales: 8.58 ± 1.49%; Eubacteriales: 0.00 ± 0.00%; Fusobacteriales: 0.12 ± 0.12%; Gastranaerophilales: 0.54 ± 0.14%; Lactobacillales: 1.62 ± 0.27%; Rhodospirillales: 0.66 ± 0.36%; Selenomonadales: 0.18 ± 0.17%; Verrucomicrobiales: 3.53 ± 1.23%).

For bacterial orders, the first four PCs collectively explained 73.40% of the variance (S2 Table). For PC1 (hereafter PC_Order1), Verrucomicrobiales contributed the most to the variance (24%), followed by Bacteroidales (20%), Fusobacteriales (19%) and Enterobacterales (14%). Verrucomicrobiales, Fusobacteriales and Enterobacterales were all positively correlated with each other, and all were negatively correlated with Bacteroidales, although the relationship was only significant between Bacteroidales and Verrucomicrobiales (S3 Table). For PC2 (hereafter PC_Order2), Selenomonadales contributed the most to the variance (24%), followed by Eubacteriales (16%), and these two bacterial orders were significantly positively correlated (S3 Table). For PC3 (hereafter PC_Order3), Gastranaerophilales contributed the most to the variance (47%), followed by Rhodospirillales (24%) and Erysipelotrichales (15%). The Gastranaerophilales were significantly positively correlated with the Erysipelotrichales, but there

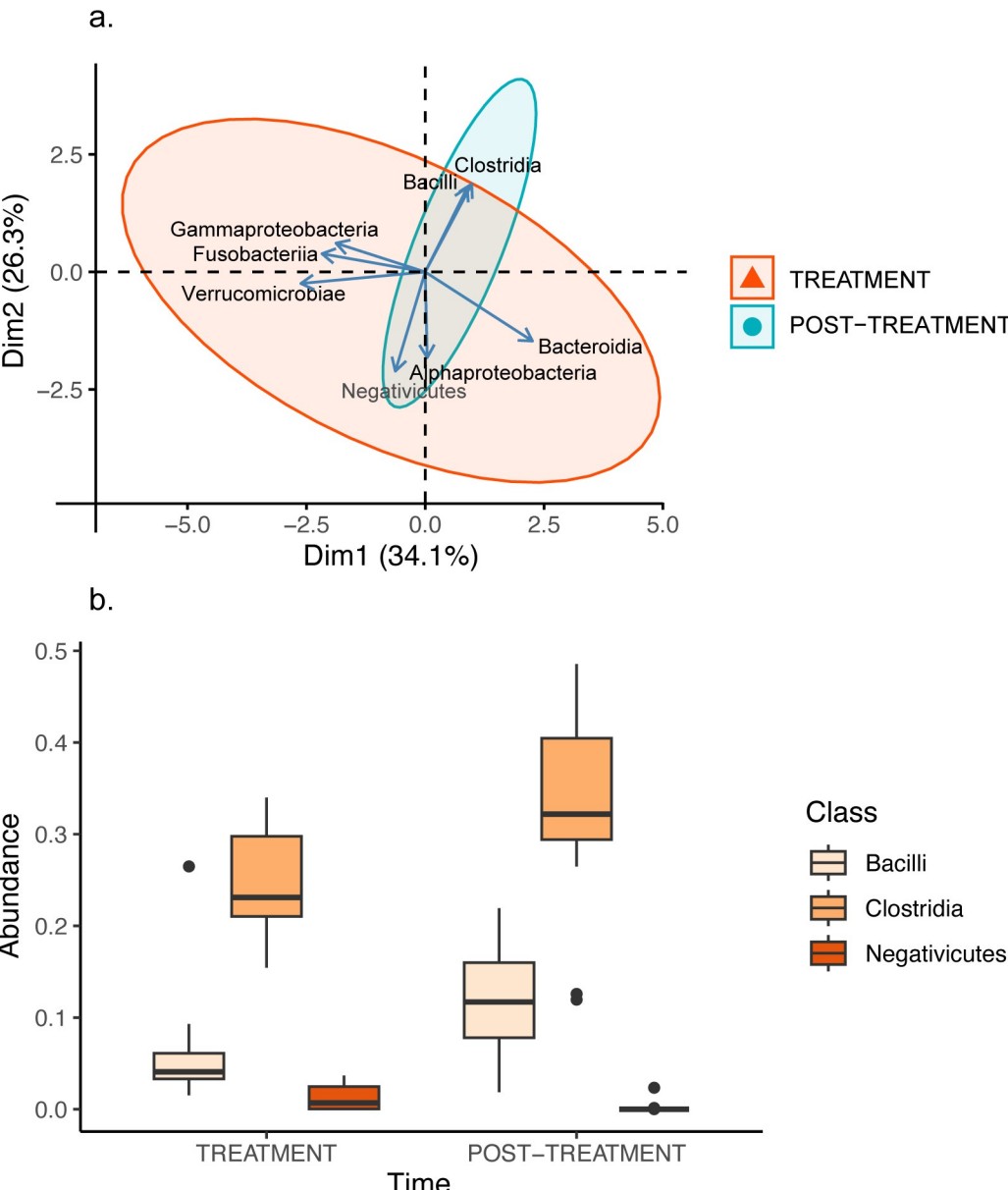

**Fig 3.** (a) Principal components analysis and (b) Box and whisker plot of three Bacterial Classes in fawn-footed mosaic-tailed rat (*Melomys cervinipes*) faecal samples. Principal components analysis shows the first two principal components of 8 bacterial classes for rats treated with antibiotics (red) and more than one year later (blue). Box and whisker plot of Class Bacilli (%), Clostridia (%) and Negativicutes (%) for both treatments.

were no other significant correlations (S3 Table). Finally, for PC4 (hereafter PC_Order4), Lactobacillales contributed the most to the variance (63%). Interestingly, the Lactobacillales were only significantly negatively correlated with Eubacteriales (S3 Table).

There was a significant treatment effect for PC_Order1 (ATS = 21.90; df = 1; $p < 0.001$; Fig 4A; $d = 1.09$ [0.28, 1.87]), which was likely associated with the abundance of Fusobacteriales (ATS = 5.62; df = 1; $p = 0.018$; $d = -0.45$ [-1.19, 0.31]) and Enterobacterales (ATS = 5.20; df = 1; $p = 0.023$; $d = -0.49$ [-1.24, 0.27]), with both groups showing significantly higher abundance during the period of antibiotic treatment and illness (Fig 4B). There was also a significant

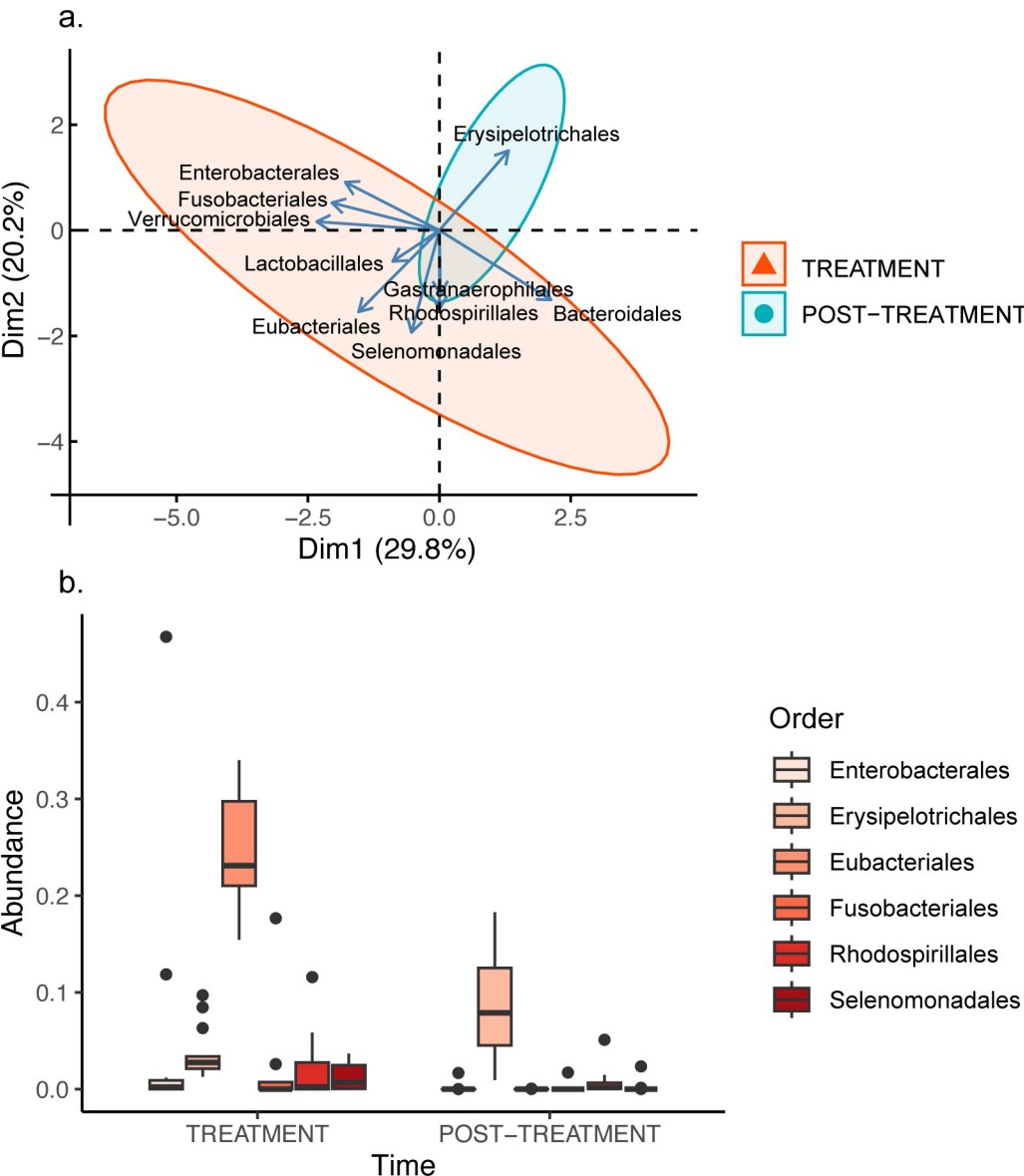

**Fig 4.** (a) Principal components analysis and (b) Box and whisker plot of six Bacterial Orders in fawn-footed mosaic-tailed rat (*Melomys cervinipes*) faecal samples. Principal components analysis shows the first two principal components of 10 bacterial orders for rats treated with antibiotics (red) and more than one year later (blue). Box and whisker plot of orders Eubacteriales (%), Erysipelotrichales (%), Rhodospirillales (%), Selenomonadales (%), Fusobacteriales (%) and Enterobacterales (%) for both treatments.

treatment effect for PC_Order2 (ATS = 46.05; df = 1; $p < 0.001$; Fig 4A; $d$ = 2.80 [1.73, 3.85]), which was likely associated with the abundance of both Eubacteriales (ATS = 210.55; df = 1; $p < 0.001$; $d$ = -1.00 [-1.78, -0.21]) and Selenomonadales (ATS = 9.68; df = 1; $p$ = 0.002; $d$ = -1.00 [-1.78, -0.21]). There was a significantly higher abundance of both bacterial orders during the period of antibiotic treatment and illness (Fig 4B). There was also a significant sex * birth * treatment interaction for PC_Order3 (ATS = 5.84; df = 1; $p$ = 0.016; Fig 4A), which was likely associated with a treatment effect (ATS = 10.27; df = 1; $p$ = 0.001; $d$ = 1.14 [0.32, 1.93]) and a birth * treatment effect (ATS = 4.70; df = 1; $p$ = 0.030) on the abundance of Erysipelotrichales and, to a lesser extent, by a near significant sex * birth * treatment effect on the

abundance of Rhodospirillales (ATS = 3.06; df = 1; $p$ = 0.080). There was a significantly lower abundance of Erysipelotrichales during the period of antibiotic treatment and illness (Fig 4B), but captive-born individuals showed a greater shift in abundance from the period of antibiotic treatment and illness to the period post-treatment from 3.5% to 10.2% compared to wild-caught individuals, which increased from 3.8% to only 6.5%. In addition, male captive-born individuals had significantly higher abundances of Rhodospirillales during the period of treatment than following treatment, and had significantly higher abundances of Rhodospirillales than male wild-caught individuals at both time periods, and female captive-born individuals in the period following treatment. There were no other significant effects (S1 Table).

## Family

49 bacterial families were common in both time periods. Of these, 17 accounted for more than 88% of the bacterial diversity, regardless of the period of time (TREATMENT: Akkermansiaceae: 8.52 ± 2.78%; Bacteroidaceae: 10.29 ± 2.59%; Clostridiales vadin BB60 group: 0.36 ± 0.34%; Enterobacteriaceae: 4.46 ± 3.36%; Erysipelotrichaceae: 3.65 ± 0.70%; Eubacteriaceae: 0.01 ± 0.00%; Fusobacteriaceae: 1.60 ± 1.25%; Lachnospiraceae: 19.42 ± 1.40%; Lactobacillaceae: 5.90 ± 1.67%; Muribaculaceae: 13.75 ± 3.55%; Peptostreptococcaceae: 2.36 ± 0.34%; Prevotellaceae: 0.37 ± 0.37%; Rhodospirillales (uncultured): 2.02 ± 0.89%; Rikenellaceae: 4.27 ± 1.49%; Oscillospiraceae: 1.85 ± 0.19%; Tannerellaceae: 13.42 ± 2.64%; Veillonellaceae: 1.21 ± 0.35%; POST-TREATMENT: Akkermansiaceae: 3.53 ± 1.23%; Bacteroidaceae: 2.24 ± 1.47%; Clostridiales vadin BB60 group: 1.31 ± 0.29%; Enterobacteriaceae: 0.12 ± 0.12%; Erysipelotrichaceae: 8.52 ± 1.51%; Eubacteriaceae: 2.43 ± 0.68%; Fusobacteriaceae: 0.12 ± 0.12%; Lachnospiraceae: 19.87 ± 1.89%; Lactobacillaceae: 1.30 ± 0.16%; Muribaculaceae: 42.00 ± 3.05%; Peptostreptococcaceae: 0.14 ± 0.12%; Prevotellaceae: 0.58 ± 0.36%; Rhodospirillales (uncultured): 0.66 ± 0.36%; Rikenellaceae: 1.08 ± 0.59%; Oscillospiraceae: 3.09 ± 0.51%; Tannerellaceae: 1.95 ± 0.86%; Veillonellaceae: 0.00 ± 0.00%).

For bacterial families, the first six PCs collectively explained 76.10% of the variance (S2 Table). For PC1 (hereafter PC_Family1), Peptostreptococcaceae contributed the most to the variance (16%), followed by Muribaculaceae (14%), Tannerellaceae (12%), Rikenellaceae (8%) and Eubacteriaceae (8%). Peptostreptococcaceae, Tannerellaceae and Rikenellaceae were all significantly positively correlated with each other, and all were negatively correlated with Muribaculaceae and Eubacteriaceae, while Muribaculaceae and Eubacteriaceae were positively correlated with each other (S3 Table). For PC2 (hereafter PC_Family2), Fusobacteriaceae contributed the most to the variance (27%), followed by Akkermansiaceae (24%) and Clostridiales vadin BB60 group (13%). Fusobacteriaceae was significantly positively correlated with Akkermansiaceae, and neither were correlated with Clostridiales vadin BB60 group (S3 Table). For PC3 (hereafter PC_Family3), Lachnospiraceae contributed the most to the variance (31%), followed by Prevotellaceae (23%). They were not significantly correlated with each other (S3 Table).

For PC4 (hereafter PC_Family4), Rhodospirillales (uncultured) contributed the most to the variance (22%), followed by Oscillospiraceae (17%). Although these two families were negatively correlated, this was not significant (S3 Table). For PC5 (hereafter PC_Family5), Enterobacteriaceae contributed the most to the variance (25%), followed by Erysipelotrichaceae (16%), Veillonellaceae (15%) and Bacteroidaceae (10%). Enterobacteriaceae and Erysipelotrichaceae were significantly negatively correlated, while Veillonellaceae and Bacteroidaceae were significantly positively correlated (S3 Table) Finally, for PC6 (hereafter PC_Family6), Lactobacillaceae contributed the most to the variance (61%).

There was a significant treatment effect for PC_Family1 (ATS = 34.41; df = 1; $p$ < 0.001; $d$ = 2.48 [1.46, 3.46]; Fig 5A), which was likely associated with the abundance of

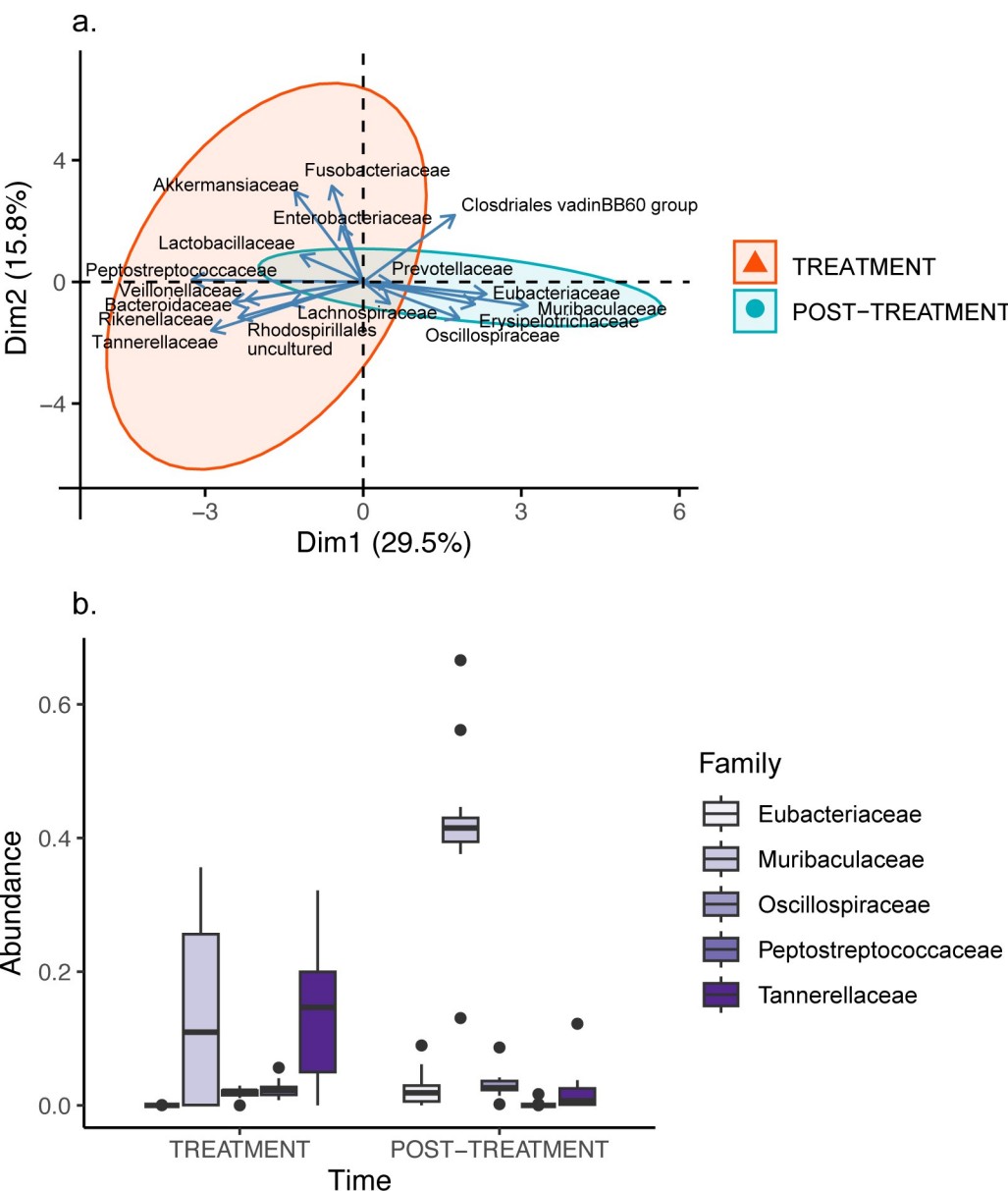

**Fig 5.** (a) Principal components analysis and (b) Box and whisker plot of 5 bacterial families in fawn-footed mosaic-tailed rat (*Melomys cervinipes*) faecal samples. Principal components analysis shows the first two principal components of 17 bacterial families for rats treated with antibiotics (red) and more than one year later (blue). Box and whisker plot of orders Eubacteriaceae (%), Muribaculaceae (%), Oscillospiraceae (%), Peptostreptococcaceae (%) and Tannerellaceae (%) for both treatments.

Eubacteriaceae (ATS = 58.76; df = 1; $p < 0.001$; $d$ = 1.35 [0.52, 2.17]), Muribaculaceae (ATS = 48.02; df = 1; $p < 0.001$; $d$ = 2.28 [1.30, 3.23]), Peptostreptococcaceae (ATS = 39.04; df = 1; $p < 0.001$; $d$ = -2.31 [-3.26, -1.33]) and Tannerellaceae (ATS = 19.17; df = 1; $p < 0.001$; $d$ = -1.56 [-2.40, -0.70]) with Eubacteriaceae and Muribaculaceae showing significantly higher abundances in the period following treatment with antibiotics and illness, and Peptostreptococcaceae and Tannerellaceae showing significantly higher abundances during the period of antibiotic treatment and illness (Fig 5B). There was also a significant treatment effect for PC_Family4 (ATS = 4.36; df = 1; $p$ = 0.037; Fig 5A; $d$ = -0.49 [-1.24, 0.27]), which

was likely associated with the abundance of Oscillospiraceae (ATS = 9.93; df = 1; *p* = 0.002; *d* = 0.86 [0.08, 1.63]), with a higher abundance being observed during the period following treatment with antibiotics and illness (Fig 5B). There were no significant treatment effects for the remaining families (S1 Table).

There was a significant sex effect for PC_Family1 (ATS = 4.00; df = 1; *p* = 0.046; *d* = -0.42 [-1.17, 0.35]), which was likely associated with sex effects for Muribaculaceae (ATS = 4.29; df = 1; *p* = 0.038; *d* = -0.42 [-1.18, 0.34]), Peptostreptococcaceae (ATS = 6.94; df = 1; *p* = 008; *d* = 0.46 [-0.30, 1.22]) and Rikenellaceae (ATS = 15.25; df = 1; *p* < 0.001; *d* = -0.75 [-1.51, 0.02]), with males having a significantly higher abundance of Muribaculaceae than females, but females having a significantly higher abundance of both Peptostreptococcaceae and Rikenellaceae. There was also a significant sex effect for PC_Family2 (ATS = 6.12; df = 1; *p* = 0.013; *d* = 0.16 [-0.58, 0.90]), although which family was driving this sex effect for this PC is not clear. There was also a significant sex * birth * treatment interaction for PC_Family4 (ATS = 4.18; df = 1; *p* = 0.041), which was likely associated with a treatment effect observed for Oscillospiraceae, and a near significant sex * birth * treatment effect for Rhodospirillales (uncultured) (ATS = 3.06; df = 1; *p* = 0.080). Wild-caught females and captive-born males showed a greater abundance of Rhodospirillales (uncultured) during the period of antibiotic treatment and illness, while the lowest abundance of Rhodospirillales (uncultured) was observed in wild-caught males during treatment and captive-born males following treatment (0.26% for each). There were no other significant effects (S1 Table).

## Bacterial community diversity and species of interest

Mean observed overall abundance was significantly higher for the period following treatment with antibiotics and illness ($t_{13}$ = -16.21, p < 0.001; Fig 6). All alpha diversity indices calculated were greater for the period following treatment with antibiotics and illness (Fig 6).

Two bacterial species from Class Gammaproteobacteria, namely *Pseudomonas aeruginosa* (Family Pseudomonadaceae) and *Stenotrophomonas maltophilia* (Family Xanthomonadaceae), were identified from faecal samples collected during treatment with antibiotics and illness, but not from samples taken a year later. Using KEGG, three biological pathways for *Pseudomonas aeruginosa* were identified (biofilm formation, exopolysaccharide biosynthesis and quorum sensing), but no pathways were identified for *Stenotrophomonas maltophilia*. One bacterial species from Phylum Bacillota, namely *Clostridium perfringens* (Family Clostridiaceae), one bacterial species from Phylum Pseudomonadota, namely *Haemophilus influenzae* (Family Pasteurellaceae), and one bacterial species from Phylum Actinomycetota, namely *Nocardiopsis dassonvillei* (Family Nocardiopsaceae), were identified from faecal samples collected more than a year following treatment with antibiotics and illness, but not from samples taken during the illness itself. However, as both *Haemophilus influenzae* and *Nocardiopsis dassonvillei* were detected in only single, separate individual mosaic-tailed rats, their presence was unlikely a response to this illness. Using KEGG, one biological pathway for quorum sensing was identified for *Clostridium perfringens*. Finally, *Clostridioides difficile* (Family Peptostreptococcaceae) was present at both time periods, but at significantly higher abundance during the period of treatment with antibiotics and illness (V = 91, p = 0.013). No pathways were identified for *C. difficile* in the KEGG database. Several antibiotic resistance genes (FunGene database), including *arna*, *beta_intI1* and *vante* were associated with all 6 bacterial species, while some in the *aac2i* family (e.g., *aac2i* and *aac2ib*) were only common to *P. aeruginosa* and *Stenotrophomonas maltophilia*. Not all associations are noted here.

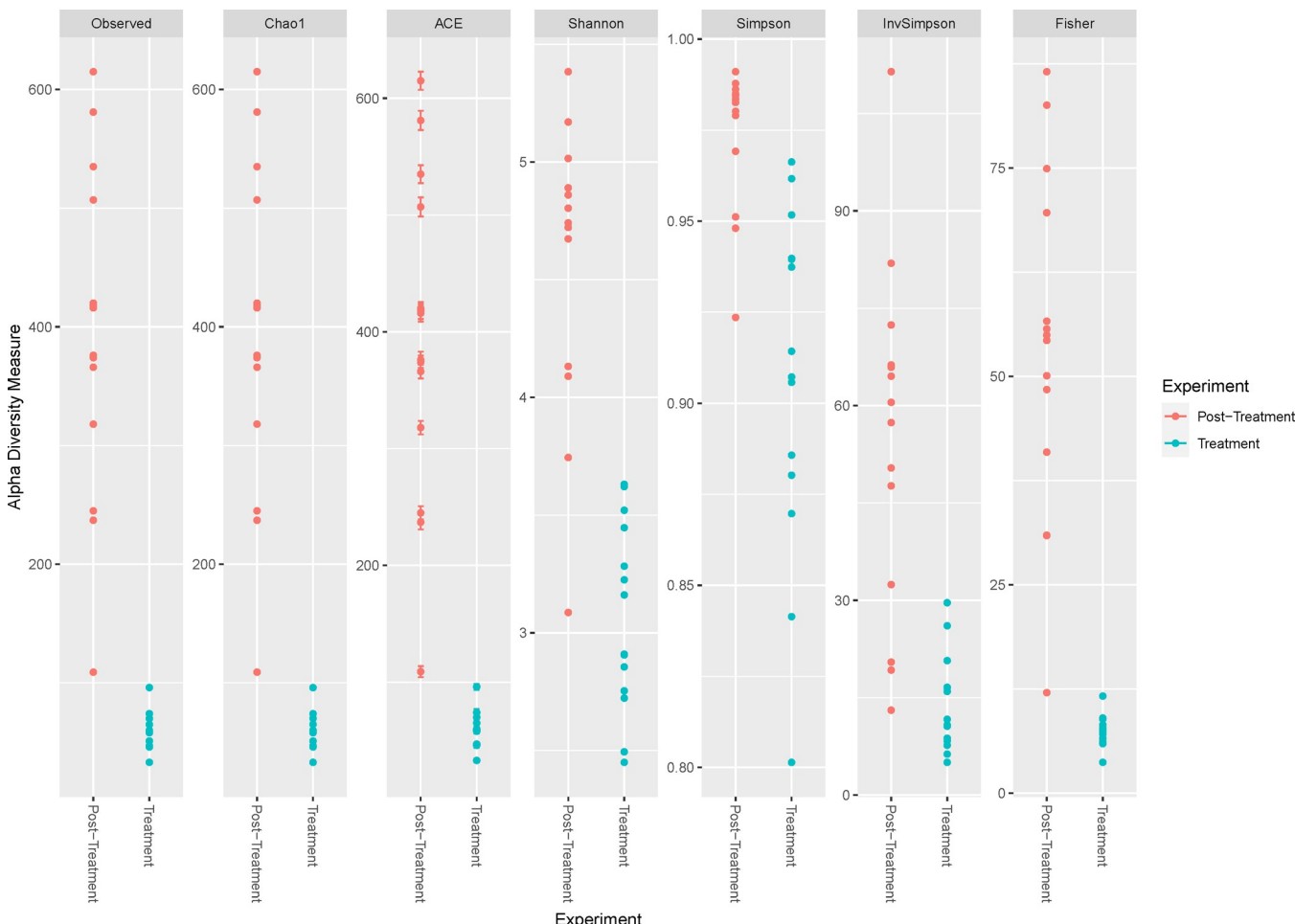

**Fig 6. Species diversity measures of the bacterial community in fawn-footed mosaic-tailed rat (*Melomys cervinipes*) faecal samples.** For rats treated with antibiotics (TREATMENT) and more than one year later (POST-TREATMENT).

## Discussion

In this study, we explored the effects of antibiotic treatment and illness on the gut microbiota of fawn-footed mosaic-tailed rats at different taxonomic levels. Because this study was opportunistic, there were no incidences of sick individuals not receiving antibiotics and later recovering, nor were there incidences of healthy animals receiving antibiotics. Therefore, the results cannot be isolated specifically to the effects of antibiotics alone, the effects of illness alone, or a combination of antibiotics and illness. Furthermore, we cannot rule out potential contamination effects due to opportunistic sampling and analysis through an external laboratory [38]. As a result, we discuss the results with a broader view to possible effects, noting that future studies will be required to clearly untangle these effects.

At the level of the domain, there was a lower abundance of Bacteria during the period of illness and treatment with antibiotics. This is not surprising, as antibiotics are commonly used to eliminate or reduce the virulence of harmful bacteria that have accumulated in the body [39]. While some antibiotics are fairly specific to their target bacteria, many antibiotics prescribed are broad-spectrum, having the capacity to affect both harmful and beneficial bacteria [40]. Furthermore, symptoms of illness, such as diarrhoea, are also known to purge the gut of microbiota [41].

The Bacteroidota and Bacillota dominated the bacterial diversity, regardless of the period of time. This is consistent with other mammals (e.g., redfronted lemurs (*Eulemur rufifrons*) [42]; koalas (*Phascolarctos cinereus*) [23]). The positive correlation between Verrucomicrobiota, Fusobacteriota and Pseudomonadota, and the increased abundance of these bacterial phyla during the period of illness and treatment with antibiotics, suggest possible direct interactions between bacterial phyla. Increased abundance of Pseudomonadota during the period of illness and treatment with antibiotics is thought to arise due to increasing epithelial oxygenation, disrupting anaerobiosis and leading to an expansion of facultative Pseudomonadota [43]. Thus, Pseudomonadota are thought to be good signatures of illness [44], signalling the risk of infection and inflammatory response [45] and a bloom in Pseudomonadota reflects gut dysbiosis [46]. Dramatic colonisation of the gut microbiota by Verrucomicrobiota following antibiotic treatment is also known [47, 48]. Because of the important role of Verrucomicrobiota in glucose homeostasis [49], perhaps an increase in abundance of bacteria in this phylum in response to Pseudomonadotal blooms could reflect an attempt by the host to restore gut dysbiosis. Furthermore, we also found that there was a lower abundance of Bacillota during the period of illness and treatment with antibiotics, consistent with Xavier [50]. Importantly, the negative correlation between Bacillota and Bacteroidia is typical of the responses of these two phyla to antibiotic treatment [40], and further demonstrates how the ratio of Bacillota to Bacteroidia (known as the F/B ratio) is indicative of gut dysbiosis, particularly with relevance to inflammatory bowel disease [51].

The Bacteroidia and Clostridia dominated the bacterial classes, regardless of the period of time. This is consistent with studies by Kim et al. [52], Pajarillo et al. [53] and Dong et al. [54]. The decreased abundance of Bacilli and Clostridia in the guts of mosaic-tailed rats during the period of illness and treatment with antibiotics is likely reflective of the negative state of the animals at the time. Ulcerative colitis is an inflammatory bowel disease, and a decrease in the abundance of Bacilli occurs during ulcerative colitis [55]. Similarly, inflammatory bowel disease is characterised by a decreased abundance of Clostridia [56]. Finally, an increased abundance of Negativicutes during the period of illness and treatment with antibiotics is consistent with other studies (e.g., feedlot cattle [57]).

The majority of Negativicutes are Gram-negative, obligate anaerobes, whereas Bacilli are predominantly Gram-positive, with both anaerobic and aerobic species. Furthermore, Clostridia, while also strictly anaerobic, contain both Gram-positive and Gram-negative species. The decreased abundance of Bacilli and Clostridia is likely a consequence of purging of the gut microbiota from diarrhoea [41], consistent with symptoms presented by the illness in the colony of mosaic-tailed rats. It is also possible that the combination of sulfamethoxazole-trimethoprim is more effective against Gram-positive Bacilli and Clostridia, contributing to their decreased abundance, whereas this treatment may be less effective against Gram-negative species, such as Negativicutes. These associations warrant further investigation.

The Bacteroidales, Eubacteriales, Lactobacillales and Verrucomicrobiales dominated the bacterial orders, regardless of the period of time. This is largely consistent with Maurice et al. [58] for wild wood mice (*Apodemus sylvaticus*). Our findings of an increase in abundance of Fusobacteriales in response to antibiotic treatment is consistent with those for pigs [59], and a disruption to the gut microbiota, such as via infection, is known to lead to blooms of Enterobacterales in other species [60], which likely explains the increased abundance observed here. An increase in the abundance of Eubacteriales under antibiotic treatment is consistent with female BALB/c mice treated with a combination of metronidazole, ampicillin, neomycin sulphate, vancomycin, and ceftriaxone sodium [61]. Interestingly, this increased abundance is suggested to represent either a compensatory response to a reduction in bacteria belonging to the family Muribaculaceae [62], which we observed here, or that bacteria in this order, being

more resistant to antibiotics [62] can flourish during these periods. However, why we found an increased abundance of Selenomonadales will require additional studies, although increased abundance of this group may be reflective of a particular diseased state (e.g., increased Selenomonadales abundance is seen for type 2 diabetes [63], but not type 1 diabetes [64]). Finally, our findings of decreased Erysipelotrichales during the period of illness and anti-biotic treatment are consistent with studies on humans experiencing Crohn's inflammatory bowel disease [65, 66], while increased Erysipelotrichales in the period following antibiotic treatment is suggestive that members of this order are highly immunogenic and can flourish post-antibiotic treatment [67, 68]. We also found that wild-caught individuals showed a slower recovery of this bacterial family over time compared to captive-born individuals. This varia-tion in the overall abundance of Erysipelotrichales because of birth origin has also been observed in the endangered Amargosa vole (*Microtus californicus scirpensis*) [69]. Variation in the abundance of Erysipelotrichales, particularly *Allobaculum*, has been linked to ingestion of a high fibre and carbohydrate rich diet [70, 71]. However, these studies showed contrasting effects. If wild-caught mosaic-tailed rats retain some of their native microbial communities in response to the diet provided, this might explain why wild-caught and captive-born individu-als differed specifically in this bacterial order.

The administration of antibiotics likely had a direct effect on the abundance of Peptostrep-tococcaceae, Muribaculaceae and Tannerellaceae, as a similar increase in abundance of Peptos-treptococcaceae and Tannerellaceae, and decrease in abundance of Muribaculaceae, in different strains of laboratory mice, is known to occur in response to treatment with antibiotics (gentamycin sulphate and cefradine: Peptostreptococcaceae [72]; amoxicillin: Muribaculaceae [73]; Clindamycin: Tannerellaceae [74]). We also found sex-specific differences in gut micro-bial composition for Muribaculaceae, Peptostreptococcaceae and Rikenellaceae. A higher abundance of Muribaculaceae in males and a higher abundance of Rikenellaceae in females is consistent with studies of C57BL/6 laboratory mice [75]. Similarly, a higher abundance of Pep-tostreptococcaceae in females is consistent with studies on B6.129S wild-type mice [76].

A decreasing abundance of Oscillospiraceae, with a corresponding increase in Bacteroida-ceae, is also consistent with a diseased state, particularly signalling the onset of inflammation [77]. Interestingly, we found a decreased abundance of Eubacteriaceae during the period of treatment with antibiotics and illness, which contrasts other studies showing the opposite pat-tern in humans treated with rifaximin [78] or in dogs in response to inflammatory bowel dis-ease [79]. A decrease in abundance of this bacterial family could suggest co-adapted or synergistic interactions with Muribaculaceae, whereby effects experienced by one family affected the other. However, this remains to be tested.

Infectious agents interact with each other, and virulence can be affected by their interac-tions with other pathogens [80]. These mechanisms can include antagonisms or synergisms (e.g., quorum sensing [81]). Antagonisms or synergisms between different bacterial orders could explain why some bacterial groups increased in abundance while others decreased in abundance in response to the antibiotics and illness. However, targeted studies are needed to determine whether the relationships between the different groups are simply a consequence of their similar responses (e.g., both Eubacteriales and Lactobacillales increased in abundance, so they may be simply positively correlated because of this) or are the outcome of particular pathobiotic mechanisms [82].

Alpha diversity indices were lower during this period of illness and treatment with antibiot-ics. While diversity indices may fail in that they may not include lower abundance taxa [83], the overall abundance of bacteria was depressed during the period, suggesting that bacterial communities are compromised by illness, antibiotics, or both. During the period of illness and treatment with antibiotics, two bacterial species with pathogenic properties were identified,

namely *Pseudomonas aeruginosa* and *Stenotrophomonas maltophilia*. Neither species was identified in the microbiome of mosaic-tailed rats assessed over a year later. *Pseudomonas aeruginosa* is an opportunistic multi-drug resistant pathogen that produces redox-active phenazines (e.g., pyocyanin and phenazine 1-carboxylic acid or PCA) involved in several biological pathways, including quorum sensing [84], biofilm formation and virulence [85, 86], iron acquisition [84, 85] and exopolysaccharide biosynthesis [87]. *Pseudomonas aeruginosa* converts PCA to pyocyanin via the phenazine-modifying genes, *phzM* and *phzS* [88] and is also known to suppress host immunity by activating the DAF-2-Insulin-like signalling pathway [89].

*Stenotrophomonas maltophilia* is an uncommon bacterium, and infection is difficult to treat [90]. It is involved in biofilm formation, with *spgM*, *rmlA*, and *rpfF* genes having a close association with biofilm formation [91]. *Stenotrophomonas maltophilia* also secretes outer membrane vesicles (OMVs) that cause an inflammatory response and stimulate the expression of proinflammatory cytokine and chemokine genes, including interleukin (IL)-1β, IL-6, IL-8, tumour necrosis factor-α and monocyte chemoattractant protein-1 [92].

*Clostridium perfringens* forms a normal component of the intestinal microflora [93]. Some of its isolates show resistance to sulfamethoxazole and trimethoprim antibiotics [94]. Thus, it is surprising that it was absent from samples obtained during the period of illness and antibiotic treatment. Why this was the case is not clear. *Haemophilus influenzae* and *Nocardiopsis dassonvillei* (an opportunistic human pathogen [95]) were each detected in only single, separate individuals, suggesting that their presence was unlikely a response to this illness.

*Clostridioides difficile* was present in the gut at both time periods; however, its abundance was higher during the period of illness and treatment with antibiotics. It is well known for its antibiotic resistance [96] and for causing serious diarrhoeal infections [97], although it can also become established in the gut without signs of disease [98]. This bacterium produces both enterotoxin [99] and cytotoxin [100], glucosyltransferases that target and inactivate the Rho family of GTPases [101], which are the causative agents of diarrhoea and inflammation. *Clostridioides difficile* is also involved in several biological pathways, including quorum sensing [102], exopolysaccharide biosynthesis, encoded by the *slpA* gene and 11 of its paralogs [103] and biofilm formation, a key regulator of which is the Spo0A gene [104].

Our results provide a greater understanding of how illness and antibiotics impact the gut microbiome of a native Australian rodent species kept under captive conditions. While the illness remains undiagnosed, antibiotic treatment was effective in curing all affected individuals, and no reoccurrence of the illness has subsequently occurred. Following the illness and treatment with antibiotics, all individuals were visually monitored daily and weighed every two weeks to assess body condition. While some potentially pathogenic bacteria were recorded in the guts of some individual mosaic-tailed rats in the period following illness and treatment with antibiotics, their general low abundance, and no physical manifestation of symptoms in individuals carrying these bacteria suggests that presence is not necessarily indicative of potential illness, and that these bacteria may simply comprise a normal component of the gut microflora, as is known for *Clostridium perfringens* [93]. Furthermore, that the overall bacterial diversity of the mosaic-tailed rat microbiome increased over time suggests that normal gut homeostasis was restored over time and provides a good baseline for future comparisons of the gut microbiota in this population.

Preventing or managing illness in captive species is a fundamental ethical issue, and humans have a duty of care to ensure that captive species are provided treatment in the event of illness (clause 3.2.1 [28]). However, as our study shows, illness and treatment with antibiotics can have vastly different effects on the different bacterial groups found in the gut, depleting some, while causing characteristic blooms in others. Future studies on the gut microbial composition of wild fawn-footed mosaic-tailed rats will be useful for understanding how captivity

affects the microbiome independently of illness and antibiotic treatment, providing new insights into the effective management of this species and related species in captivity. Furthermore, studies with carefully considered controls and deliberate manipulation of antibiotics could equally provide insights into the specific reasons for why we observed changes in the gut microbial communities of mosaic-tailed rats.

## Supporting information

**S1 Table. Output of rank-based non-parametric analyses for longitudinal data models.** Different bacterial group abundances and the effects of treatment, sex, birth and their interactions. The * refers to results that are significant at the α = 0.05 level, and significant results are discussed in the main text.
(DOCX)

**S2 Table. Outputs of principle components analyses.** Generated from the abundance of Bacteria in fawn-footed mosaic-tailed rats (*Melomys cervinipes*) in two treatments (TREATMENT and POST-TREATMENT).
(DOCX)

**S3 Table. Spearman's rank correlation matrices.** Generated for the various principal components analyses. Significant correlations indicated in bold.
(DOCX)

**S1 File. Excel spreadsheets of raw data.**
(CSV)

## Acknowledgments

We are grateful to numerous volunteers for their assistance in maintaining the mosaic-tailed rat colony. Special thanks to Dr Misha Rowell for her care of the colony during the illness, and for the collection of faecal samples. Data are available in the S1 File.

## Author Contributions

**Conceptualization:** Tasmin L. Rymer.

**Data curation:** Tasmin L. Rymer.

**Formal analysis:** Tasmin L. Rymer.

**Funding acquisition:** Tasmin L. Rymer.

**Investigation:** Tasmin L. Rymer.

**Methodology:** Tasmin L. Rymer.

**Resources:** Tasmin L. Rymer.

**Software:** Tasmin L. Rymer.

**Visualization:** Tasmin L. Rymer, Neville Pillay.

**Writing – original draft:** Tasmin L. Rymer, Neville Pillay.

**Writing – review & editing:** Tasmin L. Rymer, Neville Pillay.

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
