## [Decision Letter · Decision Letter 0]

14 Nov 2022

PONE-D-22-28589The effects of antibiotics and illness on gut microbial composition in the fawn-footed mosaic-tailed rat (Melomys cervinipes)PLOS ONE

Dear Dr. Rymer,

Thank you for submitting your manuscript to PLOS ONE. After careful consideration, we feel that it has merit but does not fully meet PLOS ONE’s publication criteria as it currently stands. Therefore, we invite you to submit a revised version of the manuscript that addresses the points raised during the review process.

 Please go through the reviewers comments and address their concerns specially about the animal health and obvious microbial changes due to antibiotic treatment. 

We look forward to receiving your revised manuscript.

Kind regards,

Mohammad Tauqeer Alam, PhD

Academic Editor

PLOS ONE

3. As part of your revision, please complete and submit a copy of the Full ARRIVE 2.0 Guidelines checklist, a document that aims to improve experimental reporting and reproducibility of animal studies for purposes of post-publication data analysis and reproducibility: https://arriveguidelines.org/sites/arrive/files/documents/Author%20Checklist%20-%20Full.pdf Please include your completed checklist as a Supporting Information file. Note that if your paper is accepted for publication, this checklist will be published as part of your article.

“The author(s) received no specific funding for this work. TR received general funding from James Cook University, and several donors contributed to a crowd funding campaign. The funders had no role in study design, data collection and analysis, decision to publish, or preparation of the manuscript.”

“We would like to thank James Cook University College of Science and Engineering for funding this project and to all the contributors in a Pozible.com crowd-funding campaign. We are grateful to numerous volunteers for their assistance in maintaining the mosaic-tailed rat colony. Special thanks to Dr Misha Rowell for her care of the colony during the illness, and for the collection of faecal samples.”

“The author(s) received no specific funding for this work. TR received general funding from James Cook University, and several donors contributed to a crowd funding campaign. The funders had no role in study design, data collection and analysis, decision to publish, or preparation of the manuscript.”

Additional Editor Comments:

Dear Dr. Rymer,

The review process of your manuscript is now complete. The reviewers have found your manuscript interesting and it has several good results. Also, the experimental design of the study and follow-up analysis approaches were appropriate. They did however raise several important issues. Additionally, both reviewers have suggested modifying the results and conclusions sections and briefly mentioning the animal health, and discussing the obvious shift in microbial structure due to antibiotic treatments.

Given the concerns and suggestions from both reviewers, unfortunately, we can not accept your manuscript in its current form. However, if you can address the reviewer's comment then we will be happy to consider the revised manuscript.

With best regards

Reviewers' comments:

Reviewer's Responses to Questions

**Comments to the Author**

1. Is the manuscript technically sound, and do the data support the conclusions?

Reviewer #1: Yes

Reviewer #2: No

2. Has the statistical analysis been performed appropriately and rigorously? 

Reviewer #1: Yes

Reviewer #2: Yes

3. Have the authors made all data underlying the findings in their manuscript fully available?

Reviewer #1: Yes

Reviewer #2: Yes

4. Is the manuscript presented in an intelligible fashion and written in standard English?

Reviewer #1: Yes

Reviewer #2: Yes

5. Review Comments to the Author

Reviewer #1: The paper described and quantified gut bacterial community composition changes in response to illness and treatment with antibiotics in a native Australian rodent, the fawn-footed mosaic-tailed rat (Melomys cervinipes), mainly by high throughput genome sequencing. This study seems to be an extension of a previously published study on a mosaic-tailed rat and builds on additional data on the dynamics of gut bacterial community composition in response to illness and treatment with antibiotics, and it is adequately planned, executed, and well-presented. NGS data could have been explored more with much deeper insight into the reason for changes in the gut bacterial community. In addition, it would have been interesting to include antibiotic-resistant bacteria (ARB) and genes (ARG), including functional analysis of bacterial community.

Reviewer #2: The authors summarized the effects of antibiotic treatment and 1-year post-treatment on gut microbial diversity in mosaic-tailed wild and captive rats. Overall, the manuscript is well-organized and well-written. Although the study was opportunistic having no controls and no significant rationale, my suggestion is to provide more detailed information on the overall health of the animal because of changes in gut microbial diversity. It would be worth adding some conclusions and suggestions for future studies on the basis of your observation otherwise this data is just information about different microbial species decreasing and increasing with antibiotics treatment. This is very obvious that gut microbial species change with antibiotics treatment and regained new homeostasis after some time with normal feed. It must be taken into account in the results section of the study to show the most important results that the research reached without going into excessive detail. In terms of the well-being of animals kept under captive conditions (it is one of the points highlighted in the article), authors must suggest how to manage the health of animals after antibiotic treatment according to their research results.

6. PLOS authors have the option to publish the peer review history of their article (what does this mean?). If published, this will include your full peer review and any attached files.

Reviewer #1: **Yes: **MUNAWWAR ALI KHAN

Reviewer #2: No

---

## [Author Response · Author response to Decision Letter 0]

4 Jan 2023

We have carefully considered the comments and respond to each comment below in bold. Where line numbers are provided, these refer to the line numbers in the unmarked manuscript (labelled “Manuscript”).

Responses to journal requirements:

We have followed the style requirements as requested.

We originally included an ethical note in the methods sections (L108-123). We now include a statement that no animals were sacrificed for this experiment (L121-122), and that all deaths were natural a consequence of the undiagnosed illness (L122-123). In the original manuscript, we stated that all animals received antibiotics (L137-139). No anaesthesia or analgesia was used/administered. We have added that sick individuals were transported to a veterinarian once symptoms started, but four died prior to receiving treatment (L118-120). As stated, all animals received antibiotics, which alleviated suffering (we stated on L120-121 that they all recovered completely).

3. As part of your revision, please complete and submit a copy of the Full ARRIVE 2.0 Guidelines checklist, a document that aims to improve experimental reporting and reproducibility of animal studies for purposes of post-publication data analysis and reproducibility: https://arriveguidelines.org/sites/arrive/files/documents/Author%20Checklist%20-%20Full.pdf Please include your completed checklist as a Supporting Information file. Note that if your paper is accepted for publication, this checklist will be published as part of your article.

We have completed the checklist as requested.

James Cook University provides researchers with facilities such as a laboratory, access to printing, access to the internet and research databases (as similar). This is what is meant by “general funding”. No grant or organization supported this work per se. This funding helped support the maintenance of the colony and provided some funds towards several projects. 

We stated this originally on submission. We state again: “The funders had no role in study design, data collection and analysis, decision to publish, or preparation of the manuscript.

No authors received a salary from any funders.

We stated this originally on submission. We state again: “The authors received no specific funding for this work.”

Included in the above queries as requested.

5. Please remove any funding-related text from the manuscript and let us know how you would like to update your Funding Statement. 

We have removed this information from the Acknowledgements section.

Please see below amended statement:

“The author(s) received no specific funding for this work. TR received general funding (e.g. access to a research laboratory, printing, research databases, etc.) from James Cook University, and several donors contributed to a crowd funding campaign that generated general research funding (e.g. not specific to this project, but allowed for the colony to be maintained). The funders had no role in study design, data collection and analysis, decision to publish, or preparation of the manuscript. No authors received a salary from any funders.”

Included as requested and provided in-text citations to match as provided in the guidelines.

Additional Editor Comments:

The review process of your manuscript is now complete. The reviewers have found your manuscript interesting and it has several good results. Also, the experimental design of the study and follow-up analysis approaches were appropriate. They did however raise several important issues. Additionally, both reviewers have suggested modifying the results and conclusions sections and briefly mentioning the animal health, and discussing the obvious shift in microbial structure due to antibiotic treatments.

Thank you for the comments. We have responded to each reviewers’ comments below in bold.

Reviewers' comments:

1. Is the manuscript technically sound, and do the data support the conclusions?

Reviewer #1: Yes

Reviewer #2: No

We have addressed each of the reviewers’ comments below each in bold.

5. Review Comments to the Author

Reviewer #1: The paper described and quantified gut bacterial community composition changes in response to illness and treatment with antibiotics in a native Australian rodent, the fawn-footed mosaic-tailed rat (Melomys cervinipes), mainly by high throughput genome sequencing. This study seems to be an extension of a previously published study on a mosaic-tailed rat and builds on additional data on the dynamics of gut bacterial community composition in response to illness and treatment with antibiotics, and it is adequately planned, executed, and well-presented. 

Thank you for the comments. There have been several other studies on mosaic-tailed rats in this colony. However, this is the first study to explore the gut bacterial community, so we have no prior knowledge or data as suggested.

NGS data could have been explored more with much deeper insight into the reason for changes in the gut bacterial community. 

This study was opportunistic. As we have no specific controls, we are unable to state with certainty that the responses are seeing are definitively a consequence of the antibiotic treatment, the illness itself or a combination of both. Given the broad approach taken to this study, we do not feel that we can provide more insight into the reason for the changes. Rather, we document the changes at the taxonomic level, and suggest that future studies could explore these changes with more controlled experiments. We have included a bit more information on the overall variation in biodiversity indices, as well as some specific bacterial species variations at each time period (L244-251, L486-514, L630-666).

In addition, it would have been interesting to include antibiotic-resistant bacteria (ARB) and genes (ARG), including functional analysis of bacterial community.

25 bacterial species were identified through the ACE pipeline analysis. Of these, only 6 occurred in both time periods. We discuss one of these, Clostridioides difficile (L657-666). Furthermore, 2 bacterial species were identified to only occur during the period of treatment, namely Pseudomonas aeruginosa and Stenotrophomonas maltophilia (L633-649). We provide a discussion on these two species. During the period following treatment, while we identified some potentially interesting pathogenic, antibiotic resistant bacteria, their low occurrence and presence in only a few individuals suggests no relationship to this particular incidence (L650-656). Given the nature of the study, and the fact that we cannot state definitively the cause of changes in the gut microbiota, we conducted a simple analysis using phyloseq to explore whether there were differences in biodiversity indices of species between the two time periods, and then explored the aforementioned species in more detail, looking very briefly at functional pathways and genes (L244-251; L630-666).

Reviewer #2: The authors summarized the effects of antibiotic treatment and 1-year post-treatment on gut microbial diversity in mosaic-tailed wild and captive rats. Overall, the manuscript is well-organized and well-written. Although the study was opportunistic having no controls and no significant rationale, my suggestion is to provide more detailed information on the overall health of the animal because of changes in gut microbial diversity. 

Thank you for the comments. We have included that sick animals showed weight loss (associated with inappetence and diarrhoea as already stated) but no other symptoms were observed (L93-94). We have provided some additional information in the methods on healthy animals (L97-99). We have also provided information on what occurred once symptoms were detected in the first four individuals (L119-119).

It would be worth adding some conclusions and suggestions for future studies on the basis of your observation otherwise this data is just information about different microbial species decreasing and increasing with antibiotics treatment. 

In the original manuscript, we stated the following (L685-689): “Future studies on the gut microbial composition of wild fawn-footed mosaic-tailed rats will be useful for understanding how captivity affects the microbiome independently of illness and antibiotic treatment, providing new insights into the effective management of this species in captivity.” We have now added “and related species” to this sentence. We also add (L689-691): “Furthermore, studies with carefully considered controls and deliberate manipulation of antibiotics could equally provide insights into the specific reasons for why we observed changes in the gut microbial communities of mosaic-tailed rats.”

This is very obvious that gut microbial species change with antibiotics treatment and regained new homeostasis after some time with normal feed. It must be taken into account in the results section of the study to show the most important results that the research reached without going into excessive detail.

Unfortunately, because this was an opportunistic study, we could not provide controls to definitively state whether the changes in community composition were a consequence of the illness, the antibiotics or both. Therefore, we chose to be cautious in our presentation and discussion of the results. We feel that we have dealt appropriately with the data at each taxonomic level at a level that is sufficient for this opportunistic study. However, we have added in some information on changes in abundance and more information on some specific species (L486-514, L630-666). If the reviewer can provide more specific recommendations on what they consider should be included and how, we will gladly consider this.

In terms of the well-being of animals kept under captive conditions (it is one of the points highlighted in the article), authors must suggest how to manage the health of animals after antibiotic treatment according to their research results.

We have provided some additional information as requested (L686-680).

---

## [Decision Letter · Decision Letter 1]

26 Jan 2023

The effects of antibiotics and illness on gut microbial composition in the fawn-footed mosaic-tailed rat (Melomys cervinipes)

PONE-D-22-28589R1

Dear Dr. Rymer,

We’re pleased to inform you that your manuscript has been judged scientifically suitable for publication and will be formally accepted for publication once it meets all outstanding technical requirements.

Kind regards,

Mohammad Tauqeer Alam, PhD

Academic Editor

PLOS ONE

Additional Editor Comments (optional):

Dear Dr Tasmin,

The review process of your manuscript is now complete. I am pleased to see that the reviewers are happy with the revised version of the manuscript. We would like to thank you for addressing the comments in the review process!

Kind regards

Reviewers' comments:

Reviewer's Responses to Questions

**Comments to the Author**

1. If the authors have adequately addressed your comments raised in a previous round of review and you feel that this manuscript is now acceptable for publication, you may indicate that here to bypass the “Comments to the Author” section, enter your conflict of interest statement in the “Confidential to Editor” section, and submit your "Accept" recommendation.

Reviewer #1: All comments have been addressed

Reviewer #2: (No Response)

2. Is the manuscript technically sound, and do the data support the conclusions?

Reviewer #1: Yes

Reviewer #2: Partly

3. Has the statistical analysis been performed appropriately and rigorously? 

Reviewer #1: Yes

Reviewer #2: Yes

4. Have the authors made all data underlying the findings in their manuscript fully available?

Reviewer #1: Yes

Reviewer #2: Yes

5. Is the manuscript presented in an intelligible fashion and written in standard English?

Reviewer #1: Yes

Reviewer #2: Yes

6. Review Comments to the Author

Reviewer #1: Thank you for revising the manuscript and addressing a few minor concerns in the previously submitted manuscript. I hope this study will lay the foundation for future in-depth research on similar or closely related study.

Reviewer #2: (No Response)

7. PLOS authors have the option to publish the peer review history of their article (what does this mean?). If published, this will include your full peer review and any attached files.

Reviewer #1: **Yes: **MUNAWWAR ALI KHAN

Reviewer #2: No

---

## [Editor Report · Acceptance letter]

14 Feb 2023

PONE-D-22-28589R1 

The effects of antibiotics and illness on gut microbial composition in the fawn-footed mosaic-tailed rat (Melomys cervinipes) 

Dear Dr. Rymer:

I'm pleased to inform you that your manuscript has been deemed suitable for publication in PLOS ONE. Congratulations! Your manuscript is now with our production department. 

Kind regards, 

on behalf of

Dr. Mohammad Tauqeer Alam 

Academic Editor

PLOS ONE